# Population Pharmacokinetic Model of Piperacillin in Critically Ill Patients and Describing Interethnic Variation Using External Validation

**DOI:** 10.3390/antibiotics11040434

**Published:** 2022-03-24

**Authors:** Cristina Sanches, Geisa C. S. Alves, Andras Farkas, Samuel Dutra da Silva, Whocely Victor de Castro, Farah Maria Drummond Chequer, Francisco Beraldi-Magalhães, Igor Rafael dos Santos Magalhães, André de Oliveira Baldoni, Mark D. Chatfield, Jeffrey Lipman, Jason A. Roberts, Suzanne L. Parker

**Affiliations:** 1Campus Centro Oeste, Universidade Federal de Sao Joao del Rei, Divinopolis 35501-296, Brazil; geisa.cristina@gmail.com (G.C.S.A.); samueldutradasilva@gmail.com (S.D.d.S.); whocely@ufsj.edu.br (W.V.d.C.); farahchequer@ufsj.edu.br (F.M.D.C.); andrebaldoni@ufsj.edu.br (A.d.O.B.); 2Optimum Dosing Strategies, Bloomingdale, NJ 07403, USA; motyocska@yahoo.com; 3Programa de Pós Graduação em Medicina Tropical, Universidade do Estado do Amazonas, Manaus 69040-000, Brazil; francisco.beraldi@gmail.com; 4School of Medicine, Faculdades Pequeno Príncipe, Curitiba 80230-020, Brazil; 5Faculdade de Ciências Farmacêuticas, Universidade Federal do Amazonas, Manaus 69077-000, Brazil; imagalhaes@ufam.edu.br; 6University of Queensland Centre for Clinical Research (UQCCR), Faculty of Medicine, The University of Queensland, Brisbane, QLD 4029, Australia; m.chatfield@uq.edu.au (M.D.C.); j.lipman@uq.edu.au (J.L.); j.roberts2@uq.edu.au (J.A.R.); suzanne.parker@uq.edu.au (S.L.P.); 7Departments of Pharmacy and Intensive Care Medicine, Royal Brisbane and Women’s Hospital, Brisbane, QLD 4029, Australia; 8Division of Anaesthesiology Critical Care Emergency and Pain Medicine, Nîmes University Hospital, University of Montpellier, 30900 Nîmes, France

**Keywords:** piperacillin, pharmacokinetics, critically ill, ethnic group, antimicrobial

## Abstract

Objectives: This study aimed to develop a piperacillin population PK model for critically ill Brazil-ian patients and describe interethnic variation using an external validation. Methods: Plasma samples were obtained from 24 ICU patients during the fifth day of piperacillin treatment and assayed by HPLC-UV. Population pharmacokinetic modelling was conducted using Pmetrics. Empiric dose of 4 g IV 6- and 8-hourly were simulated for 50 and 100% *f*T > MIC and the probabil-ity of target attainment (PTA) and the fractional target attainment (FTA) determined. Results: A two-compartment model was designed to describe the pharmacokinetics of critically ill Brazillian patients. Clearance and volume of distribution were (mean ± SD) 3.33 ± 1.24 L h^−1^ and 10.69 ± 4.50 L, respectively. Creatinine clearance was positively correlated with piperacillin clearance and a high creatinine clearance was associated with lower values of PTA and FTA. An external vali-dation was performed using data from two different ethnic ICU populations (*n* = 30), resulting in acceptable bias and precision. Conclusion: The primary pharmacokinetic parameters obtained from critically ill Brazilian patients were similar to those observed in studies performed in critically ill patients of other ethnicities. Based on our results, the use of dose adjustment based on creati-nine clearance is required in Brazilian patients.

## 1. Introduction

Sepsis is a syndromic response to infectious diseases and frequently a final pathway to death for patients with infection. In 2017, sepsis was responsible for 19.7% of all global deaths [1,2]. The incidence and mortality associated with sepsis varies substantially across geographical regions, with sub-Saharan Africa reporting the highest values (incidence from 640 up to 2500 per 100,000 population; mortality 25 to 65%) [2]. From a global perspective, the incidence of sepsis is relatively high in Brazil (approximately 440 per 100,000 population), with an associated mortality rate of 14 to 16% [2,3]. In a study of over 2600 patients, Machado et al. found an association between low availability of resources and adequacy of treatment to infection with mortality [3].

Critically ill patients can experience a deranged pathophysiology as a consequence of their disease state and medical interventions [4]. The application of pharmacokinetic/pharmacodynamic (PK/PD) principles to antimicrobial dosing of critically ill patients has shown benefits for improved clinical outcomes [4,5]. Furthermore, the use of dose adjustment tools based on population PK/PD models for treating critically ill patients is recommended [4]. However, most of the evidence describing pharmacokinetic (PK) alterations in critically ill patients and the PK/PD models used in clinically available dose adjustment tools come from studies performed using Caucasian patients [4,6,7,8]. Where dosing guidance has been established based on studies from non-ethnically diverse or non-ethnically related patients, there is the potential that dosing regimens could be sub-optimal when applied to a different ethnic patient group. This may directly impact the outcomes of ethnically diverse critically ill patients.

Pathophysiological alterations of critically ill patients between different ethnic groups remains unclear [9,10]. A systematic review by Tsai et al. (2015) identified that Asian, African, African/American, indigenous Australians and Hispanic patients can experience PK alterations, when compared to Caucasian patients. The authors identified that when dosing was adjusted for weight beta-lactam, antimicrobials were not subject to PK differences. However, beta-lactam antimicrobials, including piperacillin, are renally cleared and the adjustment for creatinine clearance may be required [6]. Otherwise, differences between renal function have been described among ethnic populations [11,12,13]. Equations used to predict glomerular filtration rate as Modification of Diet in Renal Disease (MDRD) and Chronic Kidney Disease Epidemiology Collaboration (CKD-EPI), which have been developed using studies where most participants are North American, European and few Hispanic, were found to be imprecise when applied to South American patient populations [14]. The same has been shown for the Cockcroft-Gault equation and the correction suggested for African Americans is not applicable to Brazilians [15]. Additionally, the review by Tsai et al. did not include a South American population in their research. The Brazilian population is characterized by an extensive admixture between Europeans, Africans and Amerindians [12,13].

To the best of our knowledge, no piperacillin population pharmacokinetic studies have been performed on critically ill Brazilian patients. This study aimed to develop a population PK model for critically ill Brazilian patients receiving piperacillin and to describe interethnic variation by performing an external validation. We also aimed to describe dosing regimens suitable for achieving therapeutic exposures for critically ill Brazilian patients.

## 2. Results

The study enrolled 24 critically ill patients. The patients’ demographic and clinical characteristics are presented in Table 1. Concentration–time data were obtained from the patients on the fifth day of the administration, with patients providing one or two plasma samples within a dosing interval.

A two-compartment model with a linear elimination from the central compartment, a linear inter-compartmental distribution between the central and peripheral compartment (∆-2*LL: −34.6; ∆AIC: −29.3; ∆BIC: −27.3), and with the inclusion of initial conditions to describe the steady-state conditions achieved prior to the dosing interval, improved the model (∆-2*LL: −59.4; ∆AIC: −54.1; ∆BIC: −51.9), compared to a one-compartment model. The inclusion of creatinine clearance (CRCL) normalized to 60 mL/min/1.73 m^2^ as covariate (∆-2LL: −220.9; ∆AIC: −214.8; ∆BIC: −215.1), by the equation CL = TVCL *(CRCL/60), significantly reduced the log likelihood ratio and improved the model fit as assessed by goodness-of-fit plots, with a population predicted plot correlation coefficient (r2) of 0.832, slope 0.89 (95% CI 0.75 to 1.04). Residual error was modelled as gamma * (1 + 0.1*concentration), value = 5. Parameter estimates are reported in Table 2. The observed versus predicted diagnostic plots, visual predictive check plot (*n* = 1000) and Bland-Altman residual plot of the final covariate model are presented in Figure 1.

Figure 2 presents the probability of target attainment on the fifth day of treatment for conventional intermittent dosing regimens of piperacillin for the pharmacokimnetic/pharmacodynamics (PK/PD) targets of 50% fT > MIC and 100% fT > MIC across a range of creatinine clearance values. Figure 2 shows that empiric doses of 4 g of piperacillin administered every 6 or 8 h can achieve the target of 50% fT > MIC for MIC values of up to 2 mg/L. In patients with a creatinine clearance below 30 mL/min/1.73 m^2^ doses of 4 g of piperacillin administered every 6 h achieved the target of 50% fT > MIC target for MIC values of 8 mg/L. A 4 g dose administered every 6 h or 8h and the goal of a target of 100% fT > MIC was only able to be achieved for pathogens with an MIC < 1 mg/L.

The fractional target attainment is presented in Table 3. Neither of the proposed therapies (4 g 6qh or 4 g 8qh) achieved the optimal FTA (>85%) for patients with a creatinine clearance higher than 130 mL/min.

An external validation was performed on the final covariate model using two non-Brazilian, critically ill patient populations. The model was applied to data obtained from a study in (A) critically ill Australian patients, and (B) critically ill Indigenous Australian patients (Figure 3) [16,17]. In the external validation a lower precision (root mean square prediction error, RMSPE) was observed when testing concentrations that were within the concentration range of the original dataset (<100 mg/L; Table 4).

## 3. Discussion

Despite potential differences in physiology, some of which have been previously described in various ethnic populations [11,12,13,14], the present study observed no piperacillin pharmacokinetic differences between ethnic groups for critically ill patients. This observation is supported by the bias results meeting our acceptance criteria (within 20%) in the external validation of our model designed using data from Brazilian critically ill patients with two pharmacokinetic models using data of Australian and indigenous Australian ethnic populations [16,17].

A two-compartment model with the inclusion of calculated creatinine clearance, using Cockcroft-Gault, as a covariate on piperacillin clearance was the best fit of the concentration–time data collected in our study. The design of our model using two-compartments is in agreement with published piperacillin pharmacokinetic studies in critically ill patients [17,18,19,20,21,22]. As piperacillin is largely cleared through the kidneys [23,24], a relationship between creatinine clearance and piperacillin clearance is plausible and this relationship has been described previously for critically ill Caucasian, Chinese and indigenous Australian patient populations [16,17,21,22,23].

The mean clearance obtained in the present study is similar to previously published values in critically ill patients. In our study, values for clearance ranged from 1.1 to 6.8 L/h and these values are similar to the range reported in other studies (3.6 to 5.6 L/h) [17,20,25]. Studies conducted in the Australian population report a higher value of clearance for piperacillin (14 to 17.1 L/h) [9,16,21,22], with these values being even higher than those reported for healthy volunteers (11.3 L/h) [26,27]. The present study includes patients with a median creatinine clearance of 60 mL/min/1.73 m^2^, whereas the studies in critically ill Australians reporting higher piperacillin clearance also report higher creatinine clearances, ranging from 92 to 122 mL/min/1.73 m^2^ [9,16,22]. Reinforcing the importance of the inclusion of creatinine clearance in the final model, Carrie et al. (2018) demonstrated that patients with augmented renal clearance were found to have a lower unbound piperacillin concentrations and, consequently, higher piperacillin clearance [28].

The external validation was performed using two models, by Udy et al. (2015) and Tsai et al. (2016) and we found observed concentrations were systematically lower than predicted concentrations and imprecision was more pronounced at high piperacillin concentrations (>100 mg/L), as showed on Table 4. It is likely that this is a result of extrapolating beyond the concentration range of the data in our model, with differences between predictions and observed concentrations being larger for higher piperacillin concentrations.

The central volume of piperacillin in all the studies with critically ill patients ranged from 6.8 to 19.9 L [9,16,17,20,25,28,29,30,31] and these are in the same range as healthy volunteers [27]. One exception is for the study by Alobaid et al. (2017), where a central volume of 49.0 L was reported in a study including both obese and non-obese patients. This study shows obese patients had significant higher values of central volume than non-obese patients and it is likely that this has resulted in the central volume reported, which has been calculated to include both patient groups.

In the present study, dosing simulations showed that empiric intermittent dosing of 4 g of piperacillin every 6 or 8 h was insufficient in patients with creatinine clearance >130 mL/min to achieve 50 and 100% *f*T > MIC targets when an MIC > 2 mg/L was required. Furthermore, targets of both 50 and 100% *f*T > MIC were not attained across the range of patient creatinine clearances (30 to 130 mL/min) using the empiric doses for P. aeruginosa, with an MIC of 16 mg/L. Other authors have reported similar results and simulated other approaches as continuous infusion, loading dose followed by continuous infusion, intermittent extended infusion and reduction in dosing frequency [20,21,25,28].

This study has limitations and these include: (a) only total concentrations of piperacillin were available for modelling, (b) plasma protein binding was assumed to be the same in all patients and set to 30%, (c) exact infusion times were not recorded as they were performed manually, and (d) there were no piperacillin concentrations above 100 mg/L; (e) additionally, the model was not powered to patients with decreased renal function and/or increased serum creatinine; (f) difficult in assessing ethnicity in Brazilian population. Future studies could include additional sampling in order to provide better definition of the peripherical compartment and this may reduce the variability observed for KPC. However, the high variability for KPC seen in the present study is similar to other studies of critically ill patients and not unexpected as this is a heterogenous cohort of patients.

Based on our results, the use of dose adjustment based on creatinine clearance is required in Brazilian patients.

## 4. Materials and Methods

The ClinPK checklist was used to report information in this study [32].

### 4.1. Patients and Study Setting

Patient data were retrieved from a previously described prospective, randomized clinical trial [33]. The study was performed at the ICU of a medium-sized hospital in the Midwest region of the state of Minas Gerais, Brazil. Patients were eligible to be included in the study if they were aged >18 years, with a confirmed or suspected infection with indications for use of the antibiotic piperacillin/tazobactam. Exclusion criteria included: pregnant women, individuals positive for human immunodeficiency virus (HIV) or hepatitis B or C virus, patients with a known allergy to the piperacillin/tazobactam, patients who had previously been enrolled in this study, patients with serum creatinine >2 mg/dL or elevation superior to twice the baseline value. Patients enrolled in the study received an empiric dose of 4 g q8h as an intermittent infusion over approximately 30 min or using 2 g q6h or 3.3 g q4h as an individually designed optimum dosing strategy (ID-ODS) empirical dose strategy (no TDM performed) of an intermittent infusion considering the PK/PD target of 50% *f*T > MIC. After the fifth day of treatment up to three plasma samples were collected within a dosing interval, with sampling before the dose and at 1 and 3 h after the start of infusion.

### 4.2. Drug Assay

Total piperacillin plasma concentrations were measured using a previously validated method [34] in a high-pressure liquid chromatography with UV detector on a Shimadzu Prominence system coupled to a Shimadzu UV-SPD-20A detector (Shimadzu, Kyoto, Japan), over the range of 2.5–100 mg/L. Precision was 12.5% and accuracy was within 14.2% at the tested quality control piperacillin concentrations of 5, 40 and 80 mg/L and with a dilution control of 300 mg/L.

### 4.3. Population Pharmacokinetic Modelling

A population pharmacokinetic model was developed using Pmetrics version 1.5.0 (Laboratory of Applied Pharmacokinetics and Bioinformatics, Los Angeles, CA, USA) in RStudio (version 0.99.9.3) as a wrapper for R (version 3.3.1), Xcode (version 2.6.2) and the Intel Parallel Studio Fortran Compiler XE 2017. One or two compartment structural models were constructed using the nonparametric adaptive grid (NPAG) algorithms within Pmetrics. A stepwise approach was followed in the model-building process as described below:(i).Determination of the structural base model—One or two compartment structural models were tested using the concentration–time data. The elimination of piperacillin from the central compartment was modelled as a linear process, as were the intercompartmental rate constants.(ii).Selection of the best-fit statistical error model—Additive (lambda) and multiplicative (gamma) error models were tested using a polynomial equation for standard deviation as a function of observed concentration, Y. (SD = C_0_ + C_1_.Y), with observation weighting performed as error = SD.gamma or error = (SD^2^ + lambda^2^) ^0.5^.(iii).Development of covariate model: Available clinical covariates were assessed for biological plausibility and subsequently evaluated in a covariate analysis by applying stepwise linear, log, polynomial and power regression for the continuous variables. Covariates were correlated with pharmacokinetic parameters and linear model was used for categorical variables. Selected covariates that were tested on the structural model parameters include age, height, weight, sex, body mass index, creatinine, presence of sepsis, creatinine clearance by Cockcroft-Gault and by Chronic Kidney Disease Epidemiology Collaboration (CKD-EPI), score on the Simplified Acute Physiology Score (SAPS 3), of the Multiple Organ Dysfunction Score (MODS) and Sequential Organ Failure Assessment (SOFA) at the time of sampling. Inclusion in the model was governed according to the criteria described below.(iv).Model evaluation: Model evaluation was performed by diagnostic plots and statistical examination for comparison and selection of models. The first screening was conducted by visually assessing, for each run, the goodness of fit and the coefficient of determination of the linear regression of the observed and predicted plots values (R-squared closer to 1, intercept closer to 0, slope closer to 1, lowest mean bias (as weighted predicted error) and imprecision (as (SD*(weighted predicted error))2). Secondly, methods were compared by the log-likelihood ratio test (−2*LL) for the nested model, Akaike information criterion (AIC) and Bayesian information criterion (BIC); lower values were considered the best fit. Potential covariates were separately entered into the model and statistically tested; if inclusion of the covariate resulted in an improvement in the −2*LL, AIC or BIC values and an improvement of the goodness-of-fit plots, then the covariate were retained in the final model. Finally, to evaluate the internal consistency of the model predictions with the observations, normalized prediction distribution errors (NPDE) and the posterior predictive check were assessed graphically and proportion of observations between 5th and 95th simulated percentiles above 90% were considered adequate.

A Bland-Altman analysis was performed, using BlandAltmanLeh package for R, to describe the difference between observed and predicted concentrations with prediction error versus predicted concentrations plotted, for internal validation, as well as to compare our model to two other patient populations.

### 4.4. External Validation

Two external data sets from an Australian population and an Australian Indigenous population previously published by Udy et al. (2015) [16] and Tsai et al. (2016) [17] were used to perform the external validation of the model. Piperacillin concentrations obtained at the beginning of piperacillin therapy or when at a steady state were used from both studies. The study by Udy et al. (2015) was composed of 20 Australian patients receiving 4 g of piperacillin as a 20 min infusion every 6 h; each patient had 6 blood samples collected (before the dose, at the end of infusion, and at 0.67, 1, 3.5 and 6 h after the dose). While the study by Tsai et al. (2016) was composed of 10 Indigenous Australian patients receiving a 4 g infusion three times a day, with the infusion performed from 0.5 to 1 h; each patient had 20 blood samples collected in two dosing intervals.

For the external validation the Brazilian model was used as a prior. Then a Bayesian posterior simulation was performed calculating a posterior for each subject. The linear regression, the goodness of fit and the coefficient of determination for the observed and predicted concentrations in the external validation were assessed using Pmetrics. Finally, prediction errors were evaluated to outline bias (calculated as mean prediction error [MPE]), precision (root mean square prediction error (RMSPE)) and their respective 95% CI as described by Sheiner and Beal (1981) [35]. Two ranges of concentrations were tested: (a) full data from external studies; (b) just concentrations within the prior/base model. The acceptance criteria to establish model validity was set to a bias of 20%, which has also been applied in a study by Guo et al. (2019) [36]

### 4.5. Dosing Simulations

To determine the probability of target attainment (PTA), Monte Carlo simulations (*n* = 1000) were performed using the final covariate model for different dosing regimens, BMI and a range of creatinine clearances (using the Cockcroft-Gault formula) for PK targets of 50% and 100% *f*T > MIC with 30% plasma protein binding [37,38]. The dosing regimens were simulated at a steady state for creatinine clearance of 30, 60, 90 and 130 mL/min/1.73 m^2^ and 4.0 g intermittent infusion (II) over 30 min, 6 and 8 hourly, as is the usual empiric dose. The dosing infusion system was set up to achieve a PTA superior to 90% for the target 50% *f*T > MIC and avoid continuous infusion, due to the low availability of infusion pumps in Brazilian hospitals.

PTA was determined for a steady state dosing interval only. The fractional target attainment (FTA), during the steady state dosing interval, was calculated for *P. aeruginosa* based on the MIC distribution of the European Committee for Antimicrobial Susceptibility and Testing (EUCAST) database (available at www.eucast.org accessed on: 25 October 2021). FTA was calculated considering MIC distribution within the susceptibility range defined by clinical breakpoints (16 mg/L for *P. aeruginosa*). Doses were considered optimal if the FTA was greater than 85%.

## Figures and Tables

**Figure 1 antibiotics-11-00434-f001:**
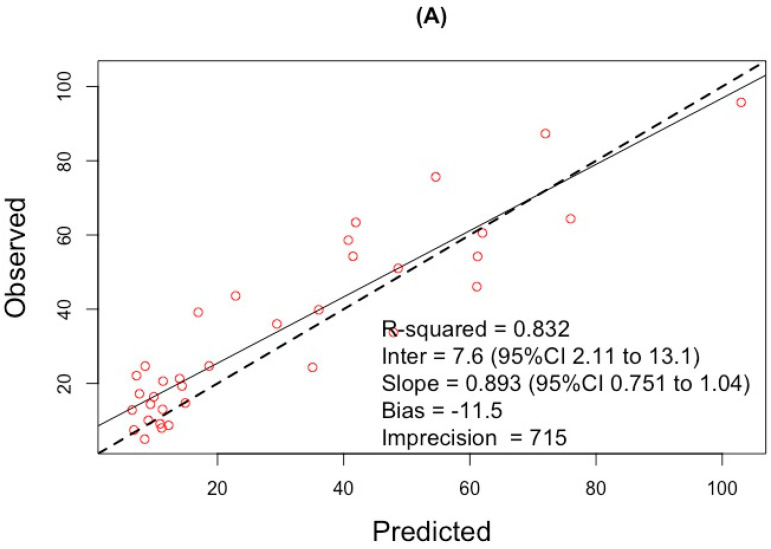
Internal validation. Observed versus population-predicted (**A**), and individual-predicted (**B**), concentration diagnostic plots, visual predictive check (**C**); (shaded areas represent quantiles distribution ranging from 0.05 to 0.95) and Bland-Altman residual plot (**D**) for individual-predicted data with a Bias of 0.01%, ULoA (+1.96SD) of 1.53% and LLoA (−1.96SD) of −1.52%.

**Figure 2 antibiotics-11-00434-f002:**
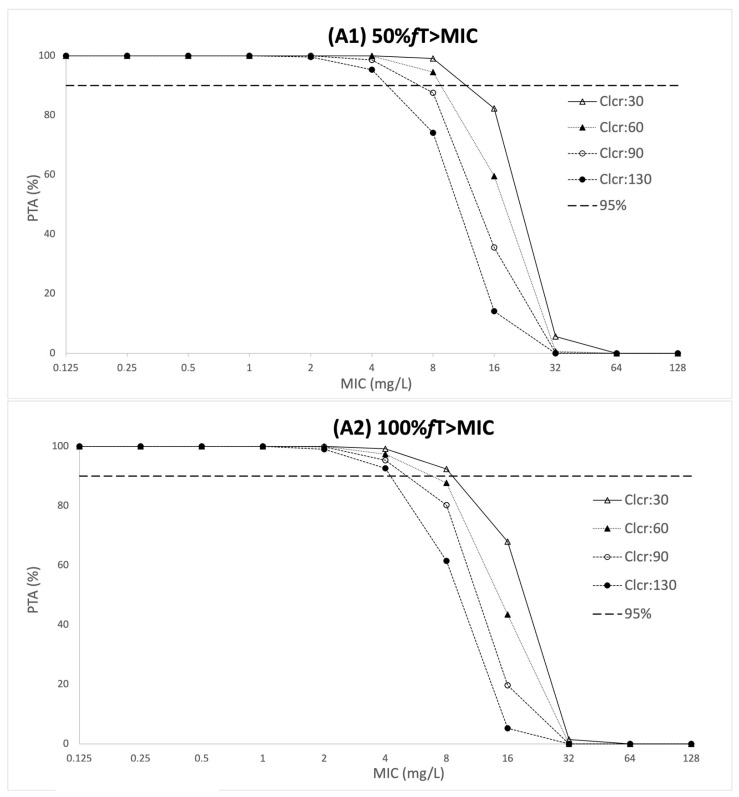
Probability of target attainment (50% *f*T > MIC and 100% *f*T > MIC) for conventional piperacillin intermittent dosing regimen: (**A**) 4 g 8-hourly, (**B**) 4g 6-hourly and intermittent infusion of 0.5 h. CLcr, creatinine clearance in mL/min/1.73 m^2^.

**Figure 3 antibiotics-11-00434-f003:**
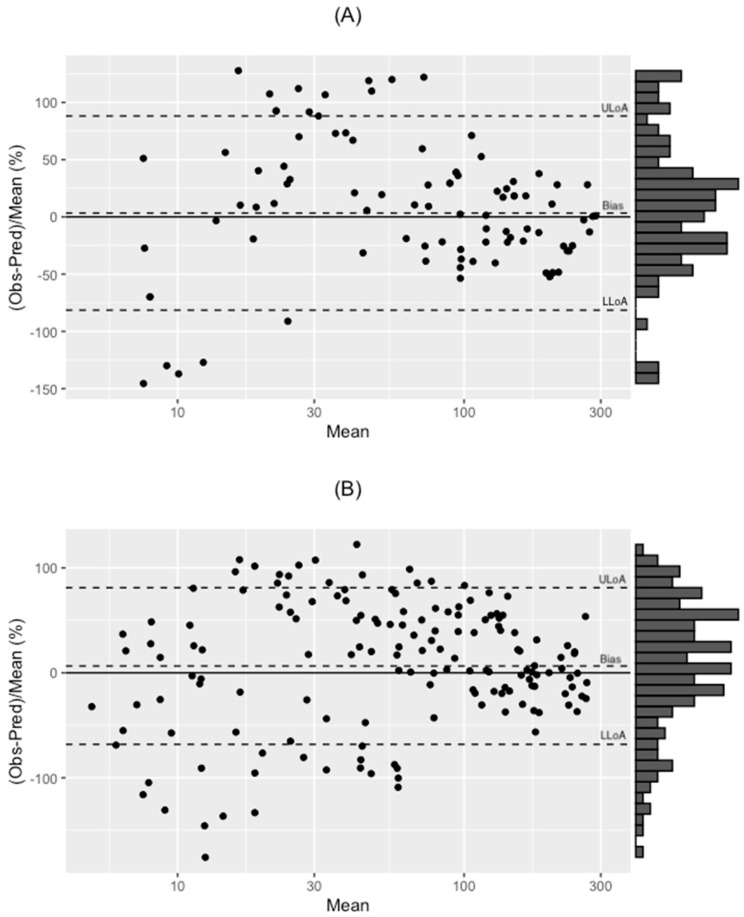
External validation plots from: (**A**) Udy et al. [16], Bias 3.3% ULoA (+1.96SD) of 88% and LLoA (−1.96SD) of −81%, and (**B**) Tsai et al. [17], Bias 6.5% ULoA (+1.96SD) of 81% and LLoA (−1.96SD) of −68%. Bland-Altman plot.

**Table 1 antibiotics-11-00434-t001:** Brazilian ICU patients’ demographic and clinical characteristics.

Characteristic	Results (*n* = 24)
Age (y)	72 (57–78)
Male	9 (38%)
Weight (kg)	69 (57–77)
BMI (kg/m^2^)	22 (21–31)
Creatinine clearance (mL/min/1.73 m^2^)	60 (47–83)
SAPS 3 score	53 (45–63)
SOFA score	5 (4–7)
MODS score	3 (2–4)
Outcome Death	8 (33%)
Vasoactive drugs	7 (29%)
Sepsis	12 (50%)
Microbiologically confirmed infection Isolated microorganism	14 (58%)
*Proteus mirabilis*	1 (7%)
*Escherichia coli*	2 (14%)
*Enterobacter aerogenes*	1 (7%)
*Pseudomonas aeruginosa*	6 (43%)
*Staphylococcus aureus*	2 (14%)
*Staphylococcus coagulase negativo*	2 (14%)
*Acinetobacter baumanni*	2 (14%)
*Serratia*	1 (7%)

BMI: body mass index; SAPS: The Simplified Acute Physiology Score; SOFA: Sequential Organ Failure Assessment; MODS: Multiple Organ Dysfunction Score; Median and Interquartile range presented for continuous measures, *n* (%) for binary measures.

**Table 2 antibiotics-11-00434-t002:** Estimates of piperacillin pharmacokinetic parameters for the final covariate model.

Parameter	Mean (SD)	Median	%CV
CL (L/h)	3.33 (1.24)	3.01	37
V (L)	10.69 (4.50)	9.03	42
KCP (h^−1^)	1.15 (0.15)	1.21	13
KPC (h^−1^)	0.08 (0.09)	0.03	120

CL, clearance; V, volume of distribution of central compartment; KCP, rate constant for piperacillin distribution from central to peripheral compartment; KPC, rate constant for piperacillin distribution from peripheral to central compartment. SD, standard deviation; CV, coefficient of variation.

**Table 3 antibiotics-11-00434-t003:** Fractional target attainment (FTA) for two piperacillin empiric dosing regimens against the EUCAST MIC distributions *P. aeruginosa*.

Dosing Regimen	FTA (%) and creatinine clearance (mL/min/1.73 m^2^)							
	4 g 6qh				4g 8qh			
	30	60	90	130	30	60	90	130
50% *f*T > MIC	97.8	94.5	89.2	82.5	97.6	93.9	88.9	81.9
100% *f*T > MIC	94.4	89.6	84.5	77.5	94.1	89.3	83.9	76.9

Shaded area indicates optimal FTA of greater than or equal to 85%.

**Table 4 antibiotics-11-00434-t004:** Description of the predictive performance of the model.

Dataset		MPE *	RMSPE *
Full data	Udy et al. [16]	−3.3 (−11.9 to 5.3)	43.2 (35.1 to 50.0)
	Tsai et al. [17]	−6.5 (−12.4 to −0.6)	38.5 (33.0 to 43.3)
<100 mg/L	Udy et al. [16]	−4.4 (−11.1 to 2.2)	24.6 (18.8 to 29.2)
	Tsai et al. [17]	−3.0 (−7.8 to 1.8)	24.6 (20.7 to 27.9)

*** Expressed as mg/L: mean (95% CI). MPE: mean prediction error; RMSPE: root mean square prediction error.

## Data Availability

Not applicable.

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
