# Peer review of "Population Pharmacokinetic Model of Piperacillin in Critically Ill Patients and Describing Interethnic Variation Using External Validation"

_antibiotics, 2022, doi:10.3390/antibiotics11040434_

Round 1

Reviewer 1 Report

Review comments on antibiotics-1625644: Population Pharmacokinetic Model of Piperacillin in Critically ill Patients and Describing Interethnic Variation using an External Validation

The manuscript described the development of a piperacillin population PK model for critically ill Brazilian patients. The authors found that the primary pharmacokinetic parameters obtained in this study were similar to those in other ethnicities. Overall, the study was appropriately designed, and the findings were important to the field. However, the data presentation (tables and figures) in this study was below the requirement for publication in Antibiotics. There were some other points to be listed as follows. Please consider them to improve the manuscript.

  1. An introduction of piperacillin is required. The authors should include relevant information on the drug, its use in sepsis treatment, and previous studies on the population pharmacokinetics of the drug. It is important to highlight that the study is within the journal’s scope.
  2. All the figures should follow the same style. The text included should be in the same font and size for consistency.
  3. Figure 1: it is better to label the sub-figures by a, b, c, d. The texts inside each sub-figure should be increased for readability.
  4. Table 4 was mentioned before Table 3 in the main text; therefore, they should be re-numbered and re-arranged.
  5. In Table 2, the KPC was estimated with a high %CV (120%). Please include relevant discussion.
  6. Figure 2 should be revised. It was unclear. The text inside each sub-figure should be increased for readability. The marker sizes need to be increased.
  7. Tables 3 and 4 should be revised following the journal’s guidelines.
  8. Size of Figure 3 should be increased for clarity.
  9. Table 4: the references should be cited in the table.
  10. Lines 176-177: the citations should be revised following the journal’s guidelines.
  11. The study had some limitations that were recognized by the authors (lines 199 – 205). The authors should propose some future studies to resolve these issues.
  12. Please check and ensure consistency among the abbreviations. For example, in table 2, KCP and KPC were mentioned; however, in the table footnote, they were Kcp and Kpc.
  13. The authors should include the data of the HPLC-UV method for drug quantitation, including linearity, precision, and accuracy in the Supplementary materials.

Author Response

Review comments on antibiotics-1625644: Population Pharmacokinetic Model of Piperacillin in Critically ill Patients and Describing Interethnic Variation using an External Validation

The manuscript described the development of a piperacillin population PK model for critically ill Brazilian patients. The authors found that the primary pharmacokinetic parameters obtained in this study were similar to those in other ethnicities. Overall, the study was appropriately designed, and the findings were important to the field. However, the data presentation (tables and figures) in this study was below the requirement for publication in Antibiotics. There were some other points to be listed as follows. Please consider them to improve the manuscript.

Dear Reviewer,

We, the authors, appreciated your comments and considerations. They contributed to manuscript improvement.

Changes were provided and highlighted in yellow along the text.

  1. An introduction of piperacillin is required. The authors should include relevant information on the drug, its use in sepsis treatment, and previous studies on the population pharmacokinetics of the drug. It is important to highlight that the study is within the journal’s scope.

Answer: We appreciate the comments. The introduction has been rewritten and now reads: “The authors identified that when dosing was adjusted for weight beta-lactam antimicrobials were not subject to PK differences. However, beta-lactam antimicrobials, including piperacillin, are renally cleared and the adjustment for creatinine clearance may be required [6]”.

  1. All the figures should follow the same style. The text included should be in the same font and size for consistency.

Answer: We appreciate the comments. Original figures are now provided as jpeg files and with the font style and size consistent.

  1. Figure 1: it is better to label the sub-figures by a, b, c, d. The texts inside each sub-figure should be increased for readability.

Answer: We appreciate the comments. Changes were performed in the main manuscript, as requested.

  1. Table 4 was mentioned before Table 3 in the main text; therefore, they should be re-numbered and re-arranged.

Answer: We appreciate the comments. Changes were performed in main manuscript and re-arranged as requested.

  1. In Table 2, the KPC was estimated with a high %CV (120%). Please include relevant discussion.

Answer: We appreciate the comments. The phrase was included in lines 202-204 in discussing limitations as follows: Future studies could include additional sampling in order to provide better definition of the peripherical compartment and this may reduce the variability observed for KPC. However,the high variability for KPC seen in the present study is similar to other studies of critically ill patients and not unexpected as this is a heterogenous cohort of patients

  1. Figure 2 should be revised. It was unclear. The text inside each sub-figure should be increased for readability. The marker sizes need to be increased.

Answer: We appreciate the comments. Original figures files are provided in jpeg stile.

  1. Tables 3 and 4 should be revised following the journal’s guidelines.

Answer: We have prepared Table 3 and 4 according to author’s guidelines, where: All table columns should have an explanatory heading. To facilitate the copy-editing of larger tables, smaller fonts may be used, but no less than 8 pt. in size. Authors should use the Table option of Microsoft Word to create tables.

  1. Size of Figure 3 should be increased for clarity.

Answer: We appreciate the comments. Changes were performed in main manuscript as requested.

  1. Table 4: the references should be cited in the table.

Answer: We appreciate the comments. Changes have been made in table, as requested.

  1. Lines 176-177: the citations should be revised following the journal’s guidelines.

Answer: We have performed these changes in the main manuscript as requested.

  1. The study had some limitations that were recognized by the authors (lines 199 – 205). The authors should propose some future studies to resolve these issues.

Answer: the phrase was included: Future studies could include additional sampling in order to provide better definition of the peripherical compartment and this may reduce the variability observed for KPC. However,the high variability for KPC seen in the present study is similar to other studies of critically ill patients and not unexpected as this is a heterogenous cohort of patients.

  1. Please check and ensure consistency among the abbreviations. For example, in table 2, KCP and KPC were mentioned; however, in the table footnote, they were Kcp and Kpc.

Answer: We appreciate the comments. We have updated the manuscript to address these concerns.

  1. The authors should include the data of the HPLC-UV method for drug quantitation, including linearity, precision, and accuracy in the Supplementary materials.

Answer: We appreciate the comments. Additional data has now been provided on item 4.2: “over the range of 2.5 – 100 mg/L.  Precision was 12.5 % and accuracy was within 14.2 % at the tested quality control piperacillin concentrations of 5, 40, and 80 mg/L and with a dilution control of 300 mg/L”.

Reviewer 2 Report

This study described a population pharmacokinetic model of piperacillin in critically ill patients. In general, there are a number of reports on population PK of piperacillin (either alone or in combination with tazobactam). Some of these reports are recent (e.g., Welch et al., Antimicrob Agents Chemother. 2021 Oct 18;65(11):e0143821; Bue et al., Int J Infect Dis. 2020 Mar;92:133-140; Hahn et al., Microbiol Spectr. 2021 Dec 22;9(3):e0063321; Sukarnjanaset et al., J Pharmacokinet Pharmacodyn. 2019 Jun;46(3):251-261; Cojutti et al., Int J Antimicrob Agents. 2021 Oct;58(4):106408.). In these studies, creatinine clearance was used as a critical covariate to reduce the variability.

Therefore, not too much new information was added to the current findings, except for the confirmation of CLcr as a significant covariate. There is no study in a particular ethnic group, but the authors did not provide any evidence that the ethnic group factor can be another covariate to be incorporated in the model.

Some other comments.

  1. The data of covariate (Clcr) should be listed in Table 2. In addition, the equations of CL and V incorporated with covariate(s) should be presented.
  2. The two-compartment model was found to best describe the observed concentration data. However, only one volume of distribution is presented, which should be V1 and V2.
  3. The top two figures of Figure 1 should be switched (usually Left and Right).

Author Response

Dear Reviewer,

We, the authors, appreciated your comments and considerations. They contributed to manuscript improvement.

Changes were provided and highlighted in yellow along the text.

This study described a population pharmacokinetic model of piperacillin in critically ill patients. In general, there are a number of reports on population PK of piperacillin (either alone or in combination with tazobactam). Some of these reports are recent (e.g., Welch et al., Antimicrob Agents Chemother. 2021 Oct 18;65(11):e0143821; Bue et al., Int J Infect Dis. 2020 Mar;92:133-140; Hahn et al., Microbiol Spectr. 2021 Dec 22;9(3):e0063321; Sukarnjanaset et al., J Pharmacokinet Pharmacodyn. 2019 Jun;46(3):251-261; Cojutti et al., Int J Antimicrob Agents. 2021 Oct;58(4):106408.). In these studies, creatinine clearance was used as a critical covariate to reduce the variability.

Therefore, not too much new information was added to the current findings, except for the confirmation of CLcr as a significant covariate. There is no study in a particular ethnic group, but the authors did not provide any evidence that the ethnic group factor can be another covariate to be incorporated in the model.

Answer: We appreciate the comments. Except for Sukarnjanaset et al (2019) the articles cited are from renal replacement therapy, ECMO and very elderly patients that are not in the scope of the manuscript. Sukarnjanaset et al (2019) was previously cited as reference [25].

To the best of our knowledge, it is the first study using PK modeling of piperacillin in a Brazilian population and provides a guide for dose adjustment for critically ill patients in this ethnic group.

Some other comments.

  1. The data of covariate (Clcr) should be listed in Table 2. In addition, the equations of CL and V incorporated with covariate(s) should be presented.

Answer: We appreciate the comments. The equation was inserted: CL = TVCL *(CRCL/60). And Clcr was listed in Table 1.

  1. The two-compartment model was found to best describe the observed concentration data. However, only one volume of distribution is presented, which should be V1 and V2.

Answer: We appreciate the comments. According to Pmetrics model library a two-compartment model can be construct using KCP and KPC or Q and Vp (http://www.lapk.org/ModelLib.php).

  1. The top two figures of Figure 1 should be switched (usually Left and Right).

Answer: We appreciate the comments. We have performed these changes in the main manuscript, as requested by reviewer.

Round 2

Reviewer 1 Report

The authors appropriately revised the manuscript according to previous comments. There are no further concerns to consider. The manuscript can be accepted as is.

Author Response

We appreciate your review. 

Reviewer 2 Report

As shown in http://www.lapk.org/ModelLib.php, the two compartment model include the parameters of Vp, Kpc, Kcp and ke. So table 2 should list ke, not CL. Please check the model carefully.

Author Response

Dear Reviewer, We appreciate the comments. 
The use of CL or Ke is optional. The model using Vp and Q (8th model in http://www.lapk.org/ModelLib.php), per example, uses CL as primary variable and the equation: Ke = CL/V to describe the parameter.